# Codesign and development of a primary school based pathway for child anxiety screening and intervention delivery: a protocol, mixed-methods feasibility study

Victoria Williamson ![ORCID] ,[1,2] Michael Larkin,[3] Tessa Reardon,[1] Samantha Pearcey,[1] Claire Hill,[4] Paul Stallard,[5] Susan H Spence,[6] Maria Breen,[4] Ian Macdonald,[7] Obioha Ukoumunne ![ORCID] ,[8] Tamsin Ford,[9] Mara Violato,[10] Falko Sniehotta,[11] Jason Stainer,[12] Alastair Gray ![ORCID] ,[10] Paul Brown,[13] Michelle Sancho,[14] Cathy Creswell[1]

For numbered affiliations see end of article.

**Correspondence to**
Dr Victoria Williamson;
victoria.williamson@kcl.ac.uk

## ABSTRACT

**Introduction** Anxiety difficulties are among the most common mental health problems in childhood. Despite this, few children access evidence-based interventions, and school may be an ideal setting to improve children's access to treatment. This article describes the design, methods and expected data collection of the Identifying Child Anxiety Through Schools – Identification to Intervention (iCATS i2i) study, which aims to develop acceptable school-based procedures to identify and support child anxiety difficulties.

**Methods and analysis** iCATS i2i will use a mixed-methods approach to codesign and deliver a set of procedures—or 'pathway'—to improve access to evidence-based intervention for child anxiety difficulties through primary schools in England. The study will consist of four stages, initially involving in-depth interviews with parents, children, school staff and stakeholders (stage 1) to inform the development of the pathway. The pathway will then be administered in two primary schools, including screening, feedback to parents and the offer of treatment where indicated (stage 2), with participating children, parents and school staff invited to provide feedback on their experience (stages 3 and 4). Data will be analysed using Template Analysis.

**Ethics and dissemination** The iCATS i2i study was approved by the University of Oxford's Research Ethics Committee (REF R64620/RE001). It is expected that this codesign study will lead on to a future feasibility study and, if indicated, a randomised controlled trial. The findings will be disseminated in several ways, including via lay summary report, publication in academic journals and presentation at conferences. By providing information on child, parent, school staff and other stakeholder's experiences, we anticipate that the findings will inform the development of an acceptable evidence-based pathway for identification and intervention for children with anxiety difficulties in primary schools and may also inform broader approaches to screening for and treating youth mental health problems outside of clinics.

## Strengths and limitations of this study

► Focus on child anxiety difficulties, one of the most common mental health problems in childhood.
► By using a codesign approach that incorporates feedback from children, parents, school staff and stakeholders, this study will lead to the development of acceptable procedures for screening and offering treatment for child anxiety difficulties in primary schools.
► The study is limited by the use of an 'opt-in' approach to consent that could introduce participation bias.
► The primary use of online platforms for consent, screening and delivery of the cognitive–behavioural therapy intervention may exclude families who have limited access to technology or lack technical skills, although ways to facilitate the participation of those in these situations will be explored.

## INTRODUCTION

Anxiety difficulties are among the most common mental health problems in childhood (6.5% prevalence[1]), and approximately half of all anxiety difficulties emerge by the age of 11 years. Childhood anxiety difficulties are often chronic and pervasive and have an adverse effect on social, education and familial functioning. Childhood anxiety difficulties often persist into adulthood when left untreated[2] and are associated with comorbid mental health difficulties, including major depression and substance abuse.[3 4] The societal cost of a child with an anxiety difficulty is estimated to be 21 times that of a non-anxious child.[5] As such,

effective identification and treatment of anxiety difficulties in childhood is important.

Effective treatments for childhood anxiety exist. However, very few children are offered or are able to access them.[6 7] For example, previous research has shown that only 2% of preadolescent children who meet criteria for an anxiety disorder in England receive an evidence-based intervention.[7] Barriers to receiving evidence-based treatment can include problems with the identification of anxiety difficulties, concerns regarding stigma to the child or family, as well as a scarcity of trained mental health professionals and long waiting lists for specialist services.[8 9] Practically speaking, attending group or face-to-face programmes can also bring logistical barriers for parents with young families including time demands and difficulties with arranging transportation or child care.[10–12]

The vast majority of children attend and spend much of their time at school; therefore, schools are also an ideal setting to overcome many of these barriers.[12 13] However, there is not a clear set of procedures for identifying youth mental health difficulties and promoting access to evidence-based treatments in schools. Moreover, previous international studies have found mixed support for school-based screening and interventions for childhood anxiety, with some studies reporting reductions in child anxiety symptoms,[12 14] while other studies have not.[15] Furthermore, some studies have reported low uptake to school-based interventions, for reasons including parents finding screening questionnaires too time consuming, parent concerns about stigma, as well as fears that their child may become more anxious from having had to discuss their worries.[14] This highlights the need for novel approaches to promote school-based approaches to increase access to early intervention for childhood anxiety difficulties that are acceptable and well tolerated in order to to increase parent participation.

One approach often used in healthcare service design and development is 'codesign', where the knowledge and lived experiences of service users themselves are drawn on to enhance the quality and experiences of care.[16 17] Codesign aims to develop an in-depth understanding of how stakeholders and service users perceive and experience the look, feel, procedures and structures of a service.[18] By engaging stakeholders and service users in codesigning a service, this is thought to lead to better quality of care and improved service performance by highlighting individual's subjective experiences at various points in the care pathway which, in turn, may lead to improvements in health outcomes and more efficient allocation of limited healthcare resources.[19] Given the importance of early intervention, an acceptable school-based pathway that incorporates the identification of children with anxiety difficulties and promotes uptake of evidence-based intervention is urgently needed. The Identifying Child Anxiety Through Schools – Identification to Intervention (iCATS i2i) study will develop procedures to identify and support child anxiety difficulties through schools informed by a codesign approach.

This article describes the iCATS i2i codesign protocol. Data collection for this study will take place between December 2019 and December 2020. The codesigned procedures will be evaluated in a subsequent feasibility study and, if indicated, randomised control trial beginning in 2021.

## METHOD

This protocol and associated procedures were approved by the Central University Research Ethics Committee at the University of Oxford (REF R64620/RE001).

### Study design

We will apply a mixed-methods approach to codesign, produce and deliver a set of procedures—or 'pathway'—to improve access to evidence-based intervention for child anxiety difficulties through primary schools (ie, ages 5–11 years) in England. Several of the key elements of the pathway were specified in advance of the codesign work based on the existing empirical literature and parent and school staff consultation. Specifically, it was prespecified that children's anxiety difficulties would be screened using questionnaire measures, parents would receive feedback and, where indicated, a brief, parent-led online intervention for child anxiety difficulties would be offered. The delivery of online treatment directly to parents was included as it has potential to overcome many of the barriers to care described above, such as logistical issues for parents and parental concerns about negative impacts on the child of participating in treatment.[7 8]

In parallel to this research, we are working on refining measures for screening for child anxiety problems (Reardon *et al*, under review), but in the interim, we will screen using brief child, parent and teacher versions of the Spence Child Anxiety Scale (SCAS-8[20]) together with four items that assess the extent of interference in everyday life (eg, 'Do fears and worries stop you from doing things?') generated to assess the impact and chronicity of and perceived need for help for anxiety difficulties. The addition of interference items is known to improve the efficacy of similar self-report measures.[21] We will consider a child to have screened 'positive' for likely anxiety difficulties if they score above the cut-off on the SCAS-8 on the basis of any reporter (score of 7.5 for parents, 6.5 for children and 4.5 for teachers) and/or indicate that anxiety interferes at least 1 'only a little' on any of the interferance items. This interference-based cut-off score was based on feedback from the dedicated stakeholder group. The use of a screening questionnaire to determine which children may benefit from additional support with anxiety was a prespecified component of the study as this approach shows promise for increasing access to support (eg, ref 22).

The intervention to be offered is an online version of a brief therapist-guided parent-delivered

cognitive–behavioural therapy (CBT) approach for child anxiety difficulties (Online Support and Intervention (OSI) for child anxiety). OSI was originally developed for use in National Health Service (NHS) clinics and was codesigned by NHS clinicians, parents and children who had received treatment for anxiety (Hill *et al*, in preparation). This intervention was selected as it is brief, effective[23] and more cost-effective than brief face-to-face psychological therapy[24] and can be delivered by non-expert practitioners (eg, ref 25). The approach of working directly with the parent, rather than the child, also addressed particular barriers to seeking and accessing help for anxiety highlighted by parents, including the preference to be supported to manage the difficulties as a family and for the child not to be singled out.[7] The online version of this intervention involves seven online modules for parents, supported by a weekly 20 min telephone call with a children's well-being practitioner (CWP; NHS Band 5), with a follow-up telephone session 4 weeks later. Modules teach parents how to explore their child's anxious thoughts, put them to the test through facing fears and to problem solve challenges that arise. This is accompanied by a game app for the child to help motivate them to face their fears. The CWPs are postgraduate psychological therapists who have received specific (12 months) training in the delivery of brief psychological therapies for children and young people who have difficulties with anxiety, low mood and behavioural disturbance. CWPs are based within settings where they can offer rapid access to psychological therapies, often including school-based clinical services, and so are the ideal workforce to implement the approach being developed if indicated.

We will use a mixed-method codesign process to determine *how* the prespecified parts of the pathway should be presented, and by whom, and to address any important considerations to optimise accessibility of and engagement with the pathway. The codesign process will consist of four stages (see figure 1) involving initial interviews and focus groups with parents, children, school staff and stakeholders (stage 1) to inform the development of a set of procedures that will be applied in two schools. These procedures will be delivered with participating children, parents and school staff (stage 2) who will provide feedback on their experience (stage 3 and 4), including cued recall specifically on the experience of receiving feedback on whether their child experiences difficulties with anxiety. Feedback from those families who choose not to be involved in the study or dropped out will also be sought to ensure any barriers to engagement are captured (stage 4).

## Patient and public involvement (PPI)

Involvement from parents, school staff and wider stakeholders informed the development of this protocol, the prespecified elements of the pathway and will contribute throughout the delivery of the codesign project. At the protocol development stage, consultation was carried out with parents, school staff/governors, leading experts in universal screening and interventions in primary school settings and representatives from key policy and practitioner organisations. Examples of decisions that were made on the basis of this consultation include specifically focusing recruitment on children in year 4 (Y4) (age 8–9 years) on the basis that this would be a manageable time for primary schools, would allow primary schools to see the benefit and for children to benefit when managing subsequent key transitions (eg, to secondary school).

Throughout the codesign process, we will consult with stakeholders in the following ways: (1) two parents with relevant lived experience, two school leaders and one mental health lead for a charity are members of the study management group and will contribute to all decisions made at a strategic level; (2) this dedicated stakeholder group will meet to review data and to make decisions to address how to solve problems and manage potentially conflicting points of view that have arisen through the codesign process; and (3) a distinct, separate online PPI group will also be formed, made up primarily of parents. Members will be invited to join via the circulation of

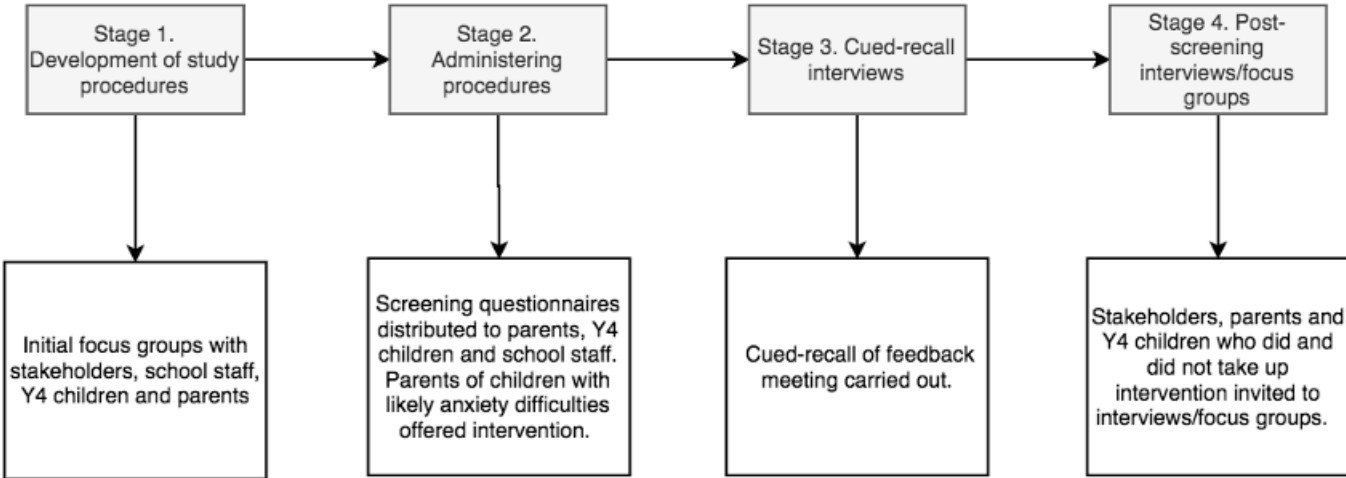

**Figure 1** . Overview of the codesign process for developing the iCATS i2i protocol. iCATS i2i, Identifying Child Anxiety Through Schools – Identification to Intervention; Y4, year 4.

**Table 1** Recruitment estimates for the codesign

| Assessment | Planned (N) | | | |
| --- | --- | --- | --- | --- |
| | **Stakeholders** | **Teachers** | **Parents** | **Children** |
| Stage 1: initial focus groups | 2 | 7 | 16 | 9 |
| Stage 2: administering procedures | | | 144 | 144 |
| Stage 3: cued recall interviews | | 4 | 12 | |
| Stage 4: postscreening interviews/focus groups | 12 | 12 | 12 | 12 |

adverts about the online group (eg, advert shared on social media and circulation of advert to parents from participating stage 2 schools), with the purpose of the group being to access wider parental views about study procedures and on key issues that arise during the study.

### Codesign participants
Participants will include Y4 children (aged 8–9 years), parents of Y4 children, primary school staff and other stakeholders. Expected participant numbers for each group and at each stage in the codesign process are outlined in table 1. These numbers are approximate, and final numbers will be informed by reviewing the range of perspectives represented in the sample and the information provided by participants.

To recruit participants with a broad range of perspectives to stage 1, we will circulate study invitations to parents of all Y4 children in two primary schools in the local Oxfordshire area, as well as using online on social media and mailing lists to recruit parents with particular experiences. For the subsequent stages, we will first contact school leaders to invite their school to participate and circulate study information to Y4 parents and children inviting them to participate.

All adult participants will be required to give written consent, and children will be required to give written assent to participate in all stages of the project.

### Inclusion criteria
Children will be eligible to participate if they are in Y4 in a mainstream primary school in England, with parent/carer consent for their participation (stages 1–4).

Parent/carers of children in Y4 in mainstream primary schools in England will be eligible to take part in stages 1–4. However, for Stage 1, we will also recruit parents through other routes in order to capture a range of experiences that might be particularly relevant to parents' engagement with and the accessibility of the pathway procedures, specifically parents who have a child with past/present mental health problem(s) or who is adopted/fostered, or where a parent has past/present mental health problem(s), or is in the military (due to their experience of frequent relocations, extended separation from parents and parental physical or psychological injuries[26]).

School staff will be included if they are employed in a mainstream primary/junior school in England (eg, class teacher and headteacher) (stages 1–4).

The inclusion criteria for wider stakeholders is that they must be a member of an organisation that is responsible for policy or practice relating to mental health provision in primary schools in England (eg, commissioning group, local authority, mental health service provider, local policy maker organisation or a governor in mainstream primary/junior schools) (stages 1 and 4).

### Procedure
We will collect data at four stages to inform the development of the pathway (see figure 1).

#### Stage 1
In this stage, we will carry out in-depth one-to-one interviews and focus groups with stakeholders, school staff, children and parents. Focus groups and interviews will draw on questioning techniques informed by the Critical Incident Approach[27] to explore participants' views about features of the pathway, which might help or hinder a positive experience or which might have been overlooked by the pathway planners altogether. Focus groups and interviews will include table-top activities where participants are shown visual representations of different aspects of the pathway, and they will be asked to discuss and write on provided notecards that will be placed on the table about their thoughts, feelings and concerns, with questions including: 'What would be the best way to do this?', 'Who do you think would be best placed to do this?', 'What might need to be done to help this part happening?', 'Where would be the best place for this to happen?', 'When is the best time to do this?' and 'Do you have any concerns about this part of the pathway?'. Photographs will be taken of the visuals produced in the focus groups and interviews for analysis. Interviews and focus groups will be audio-recorded and transcribed verbatim. These data will be used to develop a detailed prototype set of procedures for screening, feedback and intervention delivery through schools to be tested and developed further. The dedicated stakeholder group will be consulted at key decision-making points in the process to generate solutions to problems raised or inconsistent messages elicited from the interviews.

#### Stage 2
The detailed prototype set of procedures developed in stage 1 will be administered in two primary schools, including screening, feedback to parents and the offer of early intervention where indicated. We will encourage

each school to nominate a member of staff to be the 'pathway lead' (eg, a class teacher and the school's pastoral lead) who will be given training and psychoeducation by the research team about childhood anxiety difficulties and the proposed pathway procedures. The 'pathway lead' will coordinate recruitment efforts at their participating school, such as circulating study information sheets among Y4 parents. During stage 2, we will quantitatively examine pathway outcomes, including the proportion of parents in Y4 who agree to participate in the screening, the number of eligible parents who take up the intervention, the number of parents who withdraw and symptom improvement rates. We will also conduct interviews with school staff, children and parents (including children who screened 'negative' and those who screened 'positive' for anxiety and their parents, and parents who did and did not take up the intervention) to examine their experiences of the pathway and potential barriers/facilitators to engagement.

## Stage 3

On the basis of the dedicated stakeholder input at the protocol design stage, we anticipate that parents will be given written feedback on their child's screening outcomes by the school 'pathway lead', with the option of a face-to-face feedback appointment. The dedicated stakeholder group considered that feedback from the school 'pathway lead' would be preferred by families as families would likely have pre-existing relationships with the school and a member of school staff would therefore be well placed to introduce the CWP and the option to access the intervention. If this is supported by the outcomes of the earlier stages, the school staff member that is nominated to be the 'pathway lead' will receive training and guidance from the research team on delivering feedback to parents. To understand how this feedback is experienced, what works well and what parents (and the school staff 'pathway lead') find both helpful and challenging, participating parents and staff will be invited to take part in a cued recall interview meeting to allow for the refinement of future feedback delivery and staff training. To this effect, the parent–staff feedback meetings will be video recorded by the research team. Recordings of the meeting will be watched back by parents with a member of the research team to facilitate discussions about how questionnaire scores were fed back and how the opportunity to take up the intervention was shared with parents by 'pathway lead' school staff member. The researcher will invite the parent to stop the recording periodically to comment at points that are relevant to particular cues, for example, at points where the parent felt the information delivered was unhelpful or where they felt listened to. The cued recall interview will be audio-recorded and transcribed verbatim.

## Stage 4

Following the administration of the pathway procedures, interviews and focus groups will also be carried out with Y4 children, their parents and school staff in stage 4. We will carry out interviews with participating parents and children who completed the screening questionnaires and engage with the intervention modules and also with any parents and children who withdraw and parents and children who choose not to enrol in the study. School staff (eg, the 'pathway lead' and Y4 class teachers) in participating schools will be interviewed about their experience of facilitating the pathway in Y4. Relevant stakeholders will also be interviewed about their views of the pathway and how well it will fit within school settings.

### Data analysis

Focus groups and one-to-one interviews (stages 1 and 4) will be analysed using two approaches: 'fast and direct' and 'in-depth and detailed'. The 'fast and direct' analysis will use the visual outputs from focus groups and interviews to collate themes, and written summaries will provide readily understandable feedback about the pathway. Brief, complementary descriptions will be produced by following a simple protocol for verbalising 'multimodal data'[28] The combination of thematised images and verbal summary will provide immediate, easily understood feedback about the pathway.

The 'in-depth and detailed' analysis will involve Template Analysis[29] where an initial template is structured by categories drawn from relevant literature and further developed by preliminary coding of the data using a 'bottom up' approach. Once the template is developed, all transcripts will be analysed in a 'top down' manner following the provisional structure of the template. This will provide nuanced feedback about the acceptability of the pathway to fine-tune the final iteration. This analysis will capture areas of disagreement that may be missed in the 'fast and direct' analysis. Cued recall data (stage 3) will also be analysed using Template Analysis.[29] Credibility will be checked via analytic triangulation using reflective discussions with coanalysts.

### Ethics and dissemination

This research is being conducted in a community setting, and ethical approval has been obtained from the University of Oxford's Research Ethics Committee. We will seek consent from parents, school staff and other stakeholders and assent from children. Research data will be kept secure and confidential. Audio/video-recordings will require explicit consent.

A key part of this project will involve developing acceptable procedures for feeding back outcomes from screening questionnaires, which may bring potential to cause participant distress. Given that existing screening questionnaires have modest sensitivity and specificity, this includes explaining the possibility of an inaccurate result. We will pay particular attention to this throughout the codesign process to ensure we develop acceptable procedures and will seek out families who received 'false-positive' screening feedback to specifically explore their experience. Given the risk of 'false negatives', during this

codesign phase, the online intervention will be made available to all families who take part, along with information about additional resources, support and services.

## Dissemination

This project aims to develop an effective pathway to identify child anxiety difficulties in mainstream schools and deliver a parent-led intervention through ongoing collaborative work with schools, parents, children and stakeholders, while fostering avenues for disseminating the results directly to the community. Specifically, this codesign study will lead on to a future feasibility study and, if indicated, a randomised controlled trial. To disseminate the findings from this initial stage research, we will produce and disseminate a report that summarises outcomes in lay language to participating schools, which will also be shared with participating families on request. The findings will be published in high-quality, open access journals that reach both academic, educational and clinical audiences. The research team will also present the findings at national and international clinical/educational conferences. We will also collaborate with our dedicated stakeholder group to further develop the dissemination plans to ensure maximum impact.

## DISCUSSION

There is currently no evidence-based pathway for identification and intervention for children with anxiety difficulties in primary schools. Despite the existence of cost-effective psychological treatments, very few children who could benefit are able to access them.[7] This project aims to generate knowledge using a codesign approach to inform the development of such a pathway that links screening with the direct provision of support for primary school aged children to ensure that it is acceptable and ultimately implementable within schools in England. By providing information on children, parents, school staff and other stakeholder's experiences of this school-based pathway that includes screening for likely child anxiety difficulties, feedback on scores and the offer of an online intervention, we anticipate that the findings will also inform broader approaches to screening for and treating youth mental health problems outside of clinics.

This research has several methodological limitations that warrant consideration. First, because of the nature of child and parent involvement in this study, we will use an 'opt-in' approach to consent, where parents must consent to their and their child's completion of the screening measures. This is likely to introduce bias in participation, and we risk failing to capture experiences of the pathway procedures for a sufficiently broad and diverse group of families where child anxiety difficulties (including, eg, families who do not have concerns about child anxiety or where other barriers may exist, such as concerns about stigma). To address this, we will actively invite parents to stages 1 and 4 interviews who both did and did not consent to screening as well as examine in interviews whether an 'opt-out' approach to screening would be acceptable in future iterations (eg, screening measures are administered to the entire Y4 class unless parents opt-out their child from participating). A second potential limitation is that the study will primarily use online platforms for consent, and screening procedures, and to deliver the CBT intervention. This decision was informed by feedback from the dedicated stakeholder group who recommended that online participation was often considered more secure in terms of data protection and privacy. However, it introduces a risk of excluding families who have limited access to technology/Wi-Fi or lack technical skills or confidence. We will explore ways to enable participants in these situations to participate, and participant experiences of online access to the study will be examined.

With these potential limitations in mind, it is our intention that this study will collaboratively create a pathway to care for children who have problems with anxiety and their families, informed by children themselves, parents, school staff and other stakeholders, that will ultimately improve access to effective treatment and support.

**Author affiliations**

[1]Department of Experimental Psychology, Anna Watts Building, University of Oxford, Oxford, Oxfordshire, England

[2]Institute of Psychiatry, Psychology and Neuroscience, King's College London, London, England

[3]Aston Neuroscience Institute, Department of Psychology, Aston University, Birmingham, Birmingham, UK

[4]School of Psychology and Clinical Language Sciences, University of Reading, Reading, UK

[5]University of Bath, Claverton Down, Bath, UK

[6]Griffith University, 16 Russell Street South Bank, Brisbane, Queensland, Australia

[7]Charlie Waller Memorial Trust, First Floor, Rear Office, Thatcham

[8]NIHR ARC South West Peninsula, University of Exeter, Exeter, UK

[9]Department of Psychiatry, University of Cambridge, Cambridge, UK

[10]Nuffield Department of Population Health, University of Oxford, Richard Doll Building, Old Road Campus, Oxford, UK

[11]Population Health Science Institute, Faculty of Medical Sciences, Newcastle University, Baddiley-Clark Building, Richardson Road, Newcastle upon Tyne, UK

[12]Stanley Primary School, Strathmore Road, London, UK

[13]Bransgore C Of E Primary School, Ringwood Rd, Bransgore, Christchurch, UK

[14]West Berkshire Council, Council Offices, Market St, Newbury, UK

**Contributors** All authors contributed towards the design of the study protocol, contributed towards the writing of the manuscript and approved the final version of the protocol. Authors contributed towards data collection and analysis.

**Funding** Obioha Ukoumunne was supported by the National Institute for Health Research Applied Research Collaboration South West Peninsula. Mara Violato receives funding from the National Institute for Health Research (NIHR) Applied Research Collaboration Oxford and Thames Valley at Oxford Health NHS Foundation Trust, and the NIHR Oxford Biomedical Research Centre. Cathy Creswell received funding from the National Institute for Health Research (NIHR; RP-PG-0218-20010).

**Disclaimer** The views expressed are those of the authors and not necessarily those of the National Health Service, the NIHR or the Department of Health and Social Care.

**Competing interests** None declared.

**Patient and public involvement** Patients and/or the public were involved in the design, or conduct, or reporting, or dissemination plans of this research. Refer to the Methods section for further details.

**Patient consent for publication** Not required.

**Provenance and peer review** Not commissioned; externally peer reviewed.

**ORCID iDs**
Victoria Williamson http://orcid.org/0000-0002-3110-9856
Obioha Ukoumunne http://orcid.org/0000-0002-0551-9157
Alastair Gray http://orcid.org/0000-0003-0239-7278

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
