## [Reviewer comments · BMJ Open]

ARTICLE DETAILS

TITLE (PROVISIONAL)	Protocol for the co-design and development of a primary school-based pathway for child anxiety screening and intervention delivery: A mixed-methods feasibility study
AUTHORS	Williamson, Victoria; Larkin, Michael; Reardon, Tessa; Pearcey, Samantha; Hill, Claire; Stallard, Paul; Spence, Susan; Breen, Maria; Macdonald, Ian; Ukoumunne, Obioha; Ford, Tamsin; Violato, Mara; Sniehotta, Falko; Stainer, Jason; Gray, Alastair; Brown, Paul; Sancho, Michelle; Creswell, Cathy

VERSION 1 – REVIEW

REVIEWER	Bente Storm Mowatt Haugland department of Clinical Psychology, University of Bergen, Norway
REVIEW RETURNED	15-Dec-2020

GENERAL COMMENTS	This protocol paper describes the development of a school-based pathway for child anxiety screening and intervention delivery. The manuscript is very interesting, and the methods and the design described innovative (a co-design, online-program, app), appropriate, and applied in ways that is believed to increase our knowledge on important issues highly relevant for professionals within schools as well as in mental health services. As emphasized in the introduction of the manuscript, there is great need to increase the access to evidence-based interventions for children with elevated anxiety, and schools may be an ideal setting to achieve this goal. I really look forward to reading the results from this study. However, I also have some concerns about the protocol paper, and hope these may be clarified before the paper is published. 1) A central issue in this study is to improve access to evidence-based intervention for children with anxiety. Several international studies have reported on barriers to treatment for children with anxiety and their parents. It would be nice if some of the previous studies had been referred to – not only self-referral. 2) In line with the comment above – a few paragraphs should be included in the introduction describing what has been done in this area previously. What do we know already from the empirical literature on school-based interventions for children with anxiety? In what areas do we need to broaden our knowledge? 3) Hopefully, this study will bring about new insights into how to recruit and implement interventions for children with anxiety in primary schools. However, as studies have been published previously where school-based interventions have been implemented and evaluated – it is somewhat biased when the
--

	authors claim that “There is currently no evidence-based pathway for identification and intervention for children with anxiety disorders in primary schools». This relates to the same issue as in point 1 and 2. The study needs to be seen more in context of previous research. 4) The authors alternate between different terms describing the children expected to participate in the study. Sometimes the term “children with anxiety difficulties” is used whereas at other times “children with anxiety disorders” is applied (the latter is probably not correct as no diagnostic evaluation is included). This mix of concepts is confusing, and it would be helpful if they were clarified and/or applied more consistently. 5) A similar mix of terms is found when describing the intervention provided in the study. Sometimes the intervention is described as early intervention, and sometimes as treatment. It would help if this could be more congruent and the terms defined. 6) The aim of the study is “to develop an acceptable evidence-based pathway for identification and intervention for children with anxiety disorders in primary schools”. The pathway of identifying children seems to be described much more in detail– whereas the pathway of intervention is not clear. Information about this pathway should be included – or the aim of the study narrowed down. 7) The dates for the different stages in the project are lacking, as well as when the data-collection and the analyses of data are to occur. 8) It is stated that a mixed-methods approach will be applied in this study. However, when describing the analyses –qualitative data analyses only are included. I get the impression that the protocol may be a sub-study of a larger study – “the iCATS i2i”?, Perhaps the larger project might have a mixed-method design. If this is the case this should be clarified – and perhaps the mixed-method term not applied regarding this sub-study. 9) Co-design seems to be an interesting and relevant approach for this study and may have the potential of leading to better quality of care and improved service performance. However, when stating that (in the introduction page 9) this approach is “highlighting individual’s subjective feelings at various points in the care pathway”, it would be better to include some broader terms e.g., ideas, experiences, objections, to this sentence. 10) On page 10 it is stated that the study will be administered in primary schools. For readers outside of England it would be helpful if the age group for children in primary schools are given. 11) In the method section it is stated that four items are applied to assess the extent of interference from anxiety in the child’s everyday life. An example of these items could be provided – and perhaps a reference, if this has been used in previous studies: also, an explanation should be provided why one of the measures of interference from anxiety that has been tested psychometrically was not included. 12) The parents participating in the online intervention will receive support weekly from “a Children’s Wellbeing Practitioner”. It would
--	--

	be helpful to know who these practitioners are (training, competence). Why were these practitioners chosen? Are they school personnel? 13) It is not completely clear what the “online PPI group” is – who are the members of this group, and how were they selected? Is this the same group as “the dedicated stakeholders” included in the study management group? This seems to be a very important group as they are consulted at key decision making points. It would be useful to know how they were selected. 14) What is a “school mental health lead for a charity»? 15) At stage 3 – the procedure is well described – however – some questions arise about the use of school staff at this stage. School staff are expected to give feedback to parents who have children that have screened positive on the anxiety questionnaire. Which school staff are considered here? Health personnel, teachers? Do they have any training about anxiety in children or early intervention etc. before they provide this feedback? It is not altogether clear why school staff are involved at this point, why they are not included in inviting families to the intervention or in the delivery of the interventions. The role of school staff is somewhat unclear throughout the manuscript 16) Could a sentence be added to describe what “an opt-out approach” look like? 17) Some references need to be looked over. The first reference seems odd – is this a mistake? A reference is given for the statement that “most children attend school”, this seems unnecessary. However, it is relevant to add one or two references to the statement that “schools may be an ideal setting to overcome barriers towards seeking/receiving treatment”. How can school overcome these barriers? What do we know about this from previous studies? 18) Finally, a few misspellings should be corrected: Page 2+ line 5, page 5 Line 19, page 6 line 26, page 14 Line 5.
--	--

VERSION 1 – AUTHOR RESPONSE

Referee 1

1. A central issue in this study is to improve access to evidence-based intervention for children with anxiety. Several international studies have reported on barriers to treatment for children with anxiety and their parents. It would be nice if some of the previous studies had been referred to – not only self-referral.

We have added reference to the wider international literature to better highlight the difficulties families may face in accessing care, as follows (page 5):

Effective treatments for childhood anxiety exist. However, very few children are offered or are able to access them [6,7]. For example, previous research has shown that only 2% of pre-adolescent children who meet criteria for an anxiety disorder in England receive an evidence-based intervention [7]. Barriers to receiving evidence-based treatment can include problems with the identification of anxiety difficulties, concerns regarding stigma to the child or family, as well as a scarcity of trained mental health professionals and long waiting lists for specialist services [8,9]. Practically speaking, attending group or face-to-face programs can also bring logistical barriers for parents with young families including time demands, and difficulties with arranging transportation or child care [10–12].

2. In line with the comment above – a few paragraphs should be included in the introduction describing what has been done in this area previously. What do we know already from the empirical literature on school-based interventions for children with anxiety? In what areas do we need to broaden our knowledge?

We have added additional information in the introduction regarding previous research of school-based treatments for childhood anxiety, barriers to uptake and highlighted the gaps in the existing literature, as follows (page 5 & 6):

The vast majority of children attend and spend much of their time at school, therefore schools are also an ideal setting to overcome many of these barriers [10,11]. However, there is not a clear set of procedures for identifying youth mental health difficulties and promoting access to evidence-based treatments in schools. Moreover, previous international studies have found mixed support for school based screening and interventions for childhood anxiety, with some studies reporting reductions in child anxiety symptoms [10,12], while other studies have not [13]. Furthermore, some studies have reported low uptake to school-based interventions, for reasons including parents finding screening questionnaires too time consuming, parent concerns about stigma, as well as fears that their child may become more anxious from having had to discuss their worries [12]. This highlights the need for novel approaches to promote school based approaches to increase access to early intervention for childhood anxiety difficulties that are acceptable and well tolerated in order to increase parent participation.

3. Hopefully, this study will bring about new insights into how to recruit and implement interventions for children with anxiety in primary schools. However, as studies have been published previously where school-based interventions have been implemented and evaluated – it is somewhat biased when the authors claim that “There is currently no evidence-based pathway for identification and intervention for children with anxiety disorders in primary schools». This relates to the same issue as in point 1 and 2. The study needs to be seen more in context of previous research.

We have added several statements to the introduction to better present our protocol in light of the existing literature as described above (page 5-7). We also highlight that there is a need for this co-design study given that no standard of best practice currently exists in terms of procedures to screen and provide feedback to recipients to promote access to early intervention for anxiety difficulties and that new approaches are required to increase parent participation.

4. The authors alternate between different terms describing the children expected to participate in the study. Sometimes the term “children with anxiety difficulties” is used whereas at other times “children with anxiety disorders” is applied (the latter is probably not correct as no diagnostic evaluation is included). This mix of concepts is confusing, and it would be helpful if they were clarified and/or applied more consistently.

We have amended our manuscript to refer to child anxiety difficulties for consistency.

5. A similar mix of terms is found when describing the intervention provided in the study. Sometimes the intervention is described as early intervention, and sometimes as treatment. It would help if this could be more congruent and the terms defined.

We have revised our manuscript to refer to 'early intervention' throughout.

6. The aim of the study is “to develop an acceptable evidence-based pathway for identification and intervention for children with anxiety disorders in primary schools”. The pathway of identifying children seems to be described much more in detail– whereas the pathway of intervention is not clear. Information about this pathway should be included – or the aim of the study narrowed down.

We describe what the intervention to be offered to parents of children who screen 'positive' for likely anxiety difficulties consists of and we also refer to the existing literature about this online CBT intervention for readers who are interested in more information (page 8). We have not gone into further detail about the intervention as this paper is focused on developing the procedures to get families who might benefit to the intervention, rather than on developing the intervention itself. We have tried to make sure that its aim is clear throughout the manuscript.

7. The dates for the different stages in the project are lacking, as well as when the data-collection and the analyses of data are to occur.

We have added a statement (page 7) that data collection for this proposed study took place between December 2019-December 2020.

8. It is stated that a mixed-methods approach will be applied in this study. However, when describing the analyses –qualitative data analyses only are included. I get the impression that the protocol may be a sub-study of a larger study – “the iCATS i2i”?, Perhaps the larger project might have a mixed-method design. If this is the case this should be clarified – and perhaps the mixed-method term not applied regarding this sub-study.

We have more clearly described the quantitative aspects of this study, such as our aim to examine the proportion of families who participate in the school-based screening and intervention, the number of children who screen ‘positive’ for likely anxiety difficulties, the number of families who accept/reject the offer to take up the early intervention (page 13). We hope this is clearer for readers, but we are happy to make further edits if required.

9. Co-design seems to be an interesting and relevant approach for this study and may have the potential of leading to better quality of care and improved service performance. However, when stating that (in the introduction page 9) this approach is “highlighting individual’s subjective feelings at various points in the care pathway”, it would be better to include some broader terms e.g., ideas, experiences, objections, to this sentence.

We have amended our wording to refer to experiences rather than feelings. (page 6).

10. On page 10 it is stated that the study will be administered in primary schools. For readers outside of England it would be helpful if the age group for children in primary schools are given.

We have added clarification about the age group of primary school children (page 7).

11. In the method section it is stated that four items are applied to assess the extent of interference from anxiety in the child’s everyday life. An example of these items could be provided – and perhaps a reference, if this has been used in previous studies: also, an explanation should be provided why one of the measures of interference from anxiety that has been tested psychometrically was not included.

We have added an example of an interference item used in the study and clarified that the use of interference items has been found to improve the efficacy of similar self-report measures (page 7), as follows:

We will screen using brief child, parent and teacher versions of the Spence Child Anxiety Scale (SCAS-8;[18]) together with four items that assess the extent of interference in everyday life (e.g. “Do fears and worries stop you from doing things?”) generated to assess the impact and chronicity of and perceived need for help for anxiety difficulties. The addition of interference items is known to improve the efficacy of similar self-report measures [19].

12. The parents participating in the online intervention will receive support weekly from “a Children’s Wellbeing Practitioner”. It would be helpful to know who these practitioners are (training, competence). Why were these practitioners chosen? Are they school personnel?

Children’s Wellbeing Practitioners (CWPs) are postgraduate psychological therapists who have received specific (12 month) training in the delivery of brief psychological therapies for children and young people who have difficulties with anxiety, low mood, and behavioural disturbance. CWPs are based within settings where they can offer rapid access to psychological therapies, often including school based clinical services, and so are the ideal workforce to implement the approach being developed if indicated. We have added this information on page 8.

13. It is not completely clear what the “online PPI group” is – who are the members of this group, and how were they selected? Is this the same group as “the dedicated stakeholders” included in the study management group? This seems to be a very important group as they are consulted at key decision making points. It would be useful to know how they were selected.

We have provided a more detailed description to clarify the role of the online PPI group which consists of parents interested in child mental health and explained how this group differs from the dedicated stakeholder group as below (page 10),

Throughout the co-design process we will consult with stakeholders in the following ways: (i) two parents with relevant lived experience, two school leaders and one mental health lead for a charity are members of the study management group and will contribute to all decisions made at a strategic level; (ii) this dedicated stakeholder group will also meet to review data and to make decisions to address how to solve problems and manage potentially conflicting points of view that have arisen through the co-design process; and (iii) a separate online PPI group will also be formed, made up primarily of parents. Members will be invited to join via the circulation of adverts about the online

group (e.g. advert shared on social media, circulation of advert to parents from participating Stage 2 schools), with the purpose of accessing wider parental views about study procedures and on key issues that arise during the study.

14. What is a “school mental health lead for a charity”?

We have clarified that this individual leads a mental health charity (page 10).

15. At stage 3 – the procedure is well described – however – some questions arise about the use of school staff at this stage. School staff are expected to give feedback to parents who have children that have screened positive on the anxiety questionnaire. Which school staff are considered here? Health personnel, teachers? Do they have any training about anxiety in children or early intervention etc. before they provide this feedback? It is not altogether clear why school staff are involved at this point, why they are not included in inviting families to the intervention or in the delivery of the interventions. The role of school staff is somewhat unclear throughout the manuscript

We have clarified the role of the school staff and what support/training they received in delivering this feedback to parents. We describe how our plans were informed by PPI for school staff to deliver feedback and offer the intervention as it was felt that families would prefer this approach as they would likely have pre-existing relationships with schools and staff would therefore be well placed to introduce the wellbeing practitioner and intervention. We detail how the aim of the cued-recall interviews was to examine how feedback was delivered from staff to parents to determine what approaches worked well and whether there were any additional staff training needs (page 14).

On the basis of the dedicated stakeholder input at the protocol design stage, we anticipate that parents will be given written feedback on their child's screening outcomes by the school 'pathway lead', with the option of a face-to-face feedback appointment. The dedicated stakeholder group considered that feedback from the school 'pathway lead' would be preferred by families as families would likely have pre-existing relationships with the school and a member of school staff would therefore be well placed to introduce the CWP and the option to access the intervention. If this is supported by the outcomes of the earlier stages, the school staff member that is nominated to be the 'pathway lead' will receive training and guidance from the research team on delivering feedback to parents. To understand how this feedback is experienced, what works well and what parents (and 'pathway lead' school staff) find both helpful and challenging, participating parents and staff will be invited to take part in a cued-recall interview meeting to allow for the refinement of future feedback delivery and staff training.

16. Could a sentence be added to describe what “an opt-out approach” look like?

We have added a statement to clarify what an opt-out approach could look like as follows (page 17):

We will actively invite parents to Stage 1 and 4 interviews who both did and did not consent to screening as well as examine in interviews whether an ‘opt-out’ approach to screening would be acceptable in future iterations (e.g. screening measures are administered to the entire Y4 class unless parents opt-out their child from participating).

17. Some references need to be looked over. The first reference seems odd – is this a mistake?

A reference is given for the statement that “most children attend school”, this seems unnecessary. However, it is relevant to add one or two references to the statement that “schools may be an ideal setting to overcome barriers towards seeking/receiving treatment”. How can school overcome these barriers? What do we know about this from previous studies?

We thank the reviewer for bringing the problem with our references to our attention which has been rectified. We have added a number of references to both UK and international studies throughout the manuscript to ensure clarity for readers (page 5-7). We also clarify how school-based screening and intervention may overcome some existing barriers to care (please see point 2).

18. Finally, a few misspellings should be corrected: Page 2+ line 5, page 5 Line 19, page 6 line 26, page 14 Line 5.

We thank the reviewer for highlighting this and we have revised these statements.

Once again, we thank the reviewer for their time and helpful suggestions.

VERSION 2 – REVIEW

REVIEWER	Bente Storm Mowatt Haugland Department of Clinical Psychology University of Bergen
REVIEW RETURNED	19-Feb-2021
GENERAL COMMENTS	The manuscript has been very much improved from the previous version. The procedures and the different roles of participants in this interesting, but ambitious, study is now much easier to understand. I look forward to reading the results from the study!

	Please consider correcting these few point: 1) Line 17 page 3 and line 30 page 8. Is the spelling of the name of the study correct? 2) Line 49 page 7. Delete "to" 3) Line 35 page 23. Did you define the abbreviation CBT previously in the manuscript?
--	---

VERSION 2 – AUTHOR RESPONSE

Referee 1

1. Request that we address editorial lapses

We thank the reviewer for bringing these to our attention and we have made the following amendments as recommended:

Line 17 page 3 and line 30 page 8 - we have amended the typo in the study name.

Line 49 page 7. Delete "to" -we have deleted this as recommended.

Line 35 page 23. We have defined the acronym CBT in the manuscript.

Once again, we thank the reviewer for their time and helpful suggestions.